# A Quasi-Wasserstein Loss for Learning Graph Neural Networks

## ABSTRACT

When learning graph neural networks (GNNs) in node-level prediction tasks, most existing loss functions are applied for each node independently, even if node embeddings and their labels are non-i.i.d. because of their graph structures. To eliminate such inconsistency, in this study we propose a novel Quasi-Wasserstein (QW) loss with the help of the optimal transport defined on graphs, leading to new learning and prediction paradigms of GNNs. In particular, we design a "Quasi-Wasserstein" distance between the observed multi-dimensional node labels and their estimations, optimizing the label transport defined on graph edges. The estimations are parameterized by a GNN in which the optimal label transport may determine the graph edge weights optionally. By reformulating the strict constraint of the label transport to a Bregman divergence-based regularizer, we obtain the proposed Quasi-Wasserstein loss associated with two efficient solvers learning the GNN together with optimal label transport. When predicting node labels, our model combines the output of the GNN with the residual component provided by the optimal label transport, leading to a new transductive prediction paradigm. Experiments show that the proposed QW loss applies to various GNNs and helps to improve their performance in node-level classification and regression tasks.

## CCS CONCEPTS

• **Mathematics of computing** → **Graph algorithms**; • **Computing methodologies** → **Supervised learning**; **Neural networks**.

## KEYWORDS

Graph neural networks, optimal transport on graphs, Bregman divergence, transductive learning, node-level prediction

**ACM Reference Format:**
Anonymous Author(s). 2018. A Quasi-Wasserstein Loss for Learning Graph Neural Networks. In *Proceedings of Make sure to enter the correct conference title from your rights confirmation emai (Conference acronym 'XX).* ACM, New York, NY, USA, 11 pages. https://doi.org/XXXXXXX.XXXXXXX

## 1 INTRODUCTION

Graph neural network (GNN) plays a central role in many graph learning tasks, such as social network analysis [14, 35, 57], molecular modeling [24, 38, 49], transportation forecasting [29, 48], and so on. Given a graph with node features, a GNN embeds the graph nodes by exchanging and aggregating the node features, whose implementation is based on message-passing operators in the spatial

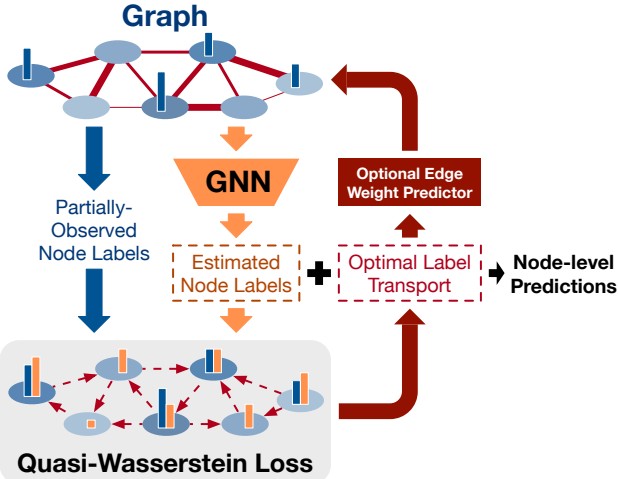

**Figure 1: The scheme of our QW-loss and the corresponding learning paradigm. Given a graph whose node features are denoted as blue circles and partially-observed node labels are denoted as blue stems, a GNN embeds the graph nodes and outputs estimated labels (denoted as orange stems). By minimizing the QW loss, we obtain the optimal label transport (denoted as the dotted red arrows on the graph edges) between the real and estimated node labels. Optionally, the optimal label transport can be used to determine the weights of graph edges (through an edge weight predictor). The final predictions are the combinations of the optimal label transport and the estimated node labels.**

domain [26, 32, 42] or graph filters in the spectral domain [4, 6, 8, 22]. When some node labels are available, we can learn the GNN in a node-level semi-supervised learning [26, 51, 56], optimizing the node embeddings to predict the observed labels. This learning framework has achieved encouraging performance in many node-level prediction tasks, e.g., node classification [31, 39].

When applying the above node-level GNN learning framework, existing work often leverages a loss function (e.g., the cross-entropy loss) to penalize the discrepancy between each node's label and the corresponding estimation. Here, some inconsistency between the objective design and the intrinsic data structure arises — the objective of learning a GNN is implemented as the summation of all the nodes' loss functions, which is often applied for i.i.d. data, but the node embeddings and their labels are non-i.i.d. in general because of the underlying graph structure and the information aggregation achieved by the GNN. As a result, the current objective treats the losses of individual nodes independently and evenly, even if the nodes in the graph are correlated and have different significance for learning the GNN. Such inconsistency may lead to sub-optimal GNNs in practice, but to our knowledge none of existing work considers this issue in-depth.

To eliminate the inconsistency, we leverage computational optimal transport techniques [34], proposing a new objective called Quasi-Wasserstein (QW) loss for learning GNNs. As illustrated in Figure 1, given partially-observed node labels and their estimations parametrized by a GNN, we consider the optimal transport between them and formulate the problem as the aggregation of the Wasserstein distances [15] corresponding to all label dimensions. This problem can be equivalently formulated as a label transport minimization problem [12, 13] defined on the graph, leading to the proposed QW loss. By minimizing this loss, we can jointly learn the optimal label transport and the GNN parametrizing the label estimations. This optimization problem can be solved efficiently by Bregman divergence-based algorithms, e.g., Bregman ADMM [45, 52]. Optionally, through a multi-layer perceptron (MLP), we can determine the edge weights of the graph based on optimal label transport, leading to a GNN with learnable edge weights.

The contributions of this study include the following two points:

- **A theoretically-solid loss without the inconsistency issue.** The QW loss provides a new optimal transport-based loss for learning GNNs, which considers the labels and estimations of graph nodes jointly. Without the questionable i.i.d. assumption, it eliminates the inconsistency issue mentioned above. In theory, we demonstrate that the QW loss is a valid metric for the node labels defined on graphs. Additionally, the traditional objective function for learning GNNs can be treated as a special case of our QW loss. We further demonstrate that applying our QW loss reduces data fitting errors in the training phase.
- **New learning and prediction paradigms.** Different from the existing methods that combine GNNs with label propagation mechanisms [9, 23, 46], the QW loss provides a new way to combine node embeddings with label information in both training and testing phases. In particular, Bregman divergence-based algorithms are applied to learn the model, and the final model consists of the GNN and a residual component provided by the optimal label transport. When predicting node labels, the model combines the estimations provided by the GNN with the complementary information from the optimal label transport, leading to a new transductive prediction paradigm.

Experiments demonstrate that our QW loss applies to various GNNs and helps to improve their performance in various node-level classification and regression tasks.

## 2 RELATED WORK

### 2.1 Graph Neural Networks

Graph neural networks can be coarsely categorized into two classes. The GNNs in the first class apply spatial convolutions to graphs [32]. The representative work includes the graph convolutional network (GCN) in [26], the graph attention network (GAT) in [42], and their variants [16, 47, 60]. The GNNs in the second class achieve graph spectral filtering [19]. They are often designed based on a polynomial basis, such as ChebNet [8] and its variants [21], GPR-GNN [6], and BernNet [22]. Besides approximated by the polynomial basis, the spectral GNNs can be learned by other strategies, e.g., Personalized PageRank in APPNP [17], graph optimization functions

in GNN-LF/HF [58], ARMA filters [4], and diffusion kernel-based filters [11, 27, 51].

The above spatial and spectral GNNs are correlated because a spatial convolution always corresponds to a graph spectral filter [2]. For example, GCN [26] can be explained as a low-pass filter achieved by a first-order Chebyshev polynomial. Given a graph with some labeled nodes, we often learn the above GNNs in a semi-supervised node-level learning framework [26, 56], in which the GNNs embed all the nodes and are trained under the supervision of the labeled nodes. However, as aforementioned, the objective functions used in the framework treat the graph nodes independently and thus mismatch with the non-i.i.d. nature of the data.

### 2.2 Computational Optimal Transport

As a powerful mathematical tool, optimal transport (OT) distance (or called Wasserstein distance under some specific settings) provides a valid metric for probability measures [43], which has been widely used for various machine learning problems, e.g., distribution comparison [15, 28], point cloud registration [18], graph partitioning [10, 54], generative modeling [1, 41], and so on. Typically, the OT distance corresponds to a constrained linear programming problem. To approximate the OT distance with low complexity, many algorithms have been proposed, e.g., Sinkhorn-scaling [7], Bregman ADMM [45], Conditional Gradient [40], and Inexact Proximal Point [50]. Recently, two iterative optimization methods have been proposed to solve the optimal transport problems defined on graphs [12, 13].

These efficient algorithms make the OT distance a feasible loss for machine learning problems, e.g., the Wasserstein loss in [15]. Focusing on the learning of GNNs, the work in [5] proposes a Wasserstein distance-based contrastive learning method. The Gromovized Wasserstein loss is applied to learn cross-graph node embeddings [54], graph factorization models [44, 52], and GNN-based graph autoencoders [53]. The above work is designed for graph-level learning tasks, e.g., graph matching, representation, classification, and clustering. Our QW loss, on the contrary, is designed for node-level prediction tasks, resulting in significantly different learning and prediction paradigms.

## 3 PROPOSED METHOD

### 3.1 Motivation and Principle

Denote a graph as $G(\mathcal{V}, \mathcal{E})$, where $\mathcal{V}$ represents the set of nodes and $\mathcal{E}$ represents the set of edges, respectively. The graph $G$ is associated with an adjacency matrix $A \in \mathbb{R}^{|\mathcal{V}| \times |\mathcal{V}|}$ and a edge weight vector $\boldsymbol{w} = [w_e] \in \mathbb{R}^{|\mathcal{E}|}$. The weights in $\boldsymbol{w}$ correspond to the non-zero elements in $A$. For an unweighted graph, $A$ is a binary matrix, and $\boldsymbol{w}$ is an all-one vector. Additionally, the nodes of the graph may have $D$-dimensional features, which are formulated as a matrix $X \in \mathbb{R}^{|\mathcal{V}| \times D}$. Suppose that a subset of nodes, denoted as $\mathcal{V}_L \subset \mathcal{V}$, are annotated with $C$-dimensional labels, i.e., $\{\boldsymbol{y}_v \in \mathbb{R}^C\}_{v \in \mathcal{V}_L}$. We would like to learn a GNN to predict the labels of the remaining nodes, i.e., $\{\boldsymbol{y}_v\}_{v \in \mathcal{V} \setminus \mathcal{V}_L}$.

The motivation for applying GNNs is based on the non-i.i.d. property of the node features and labels. Suppose that we have two nodes connected by an edge, i.e., $(v, v') \in \mathcal{E}$, where $(\boldsymbol{x}_v, \boldsymbol{y}_v)$ and $(\boldsymbol{x}_{v'}, \boldsymbol{y}_{v'})$ are their node features and labels. For each node, its neighbors'

features or labels can provide valuable information to its prediction task, i.e., the conditional probability $p(\boldsymbol{y}_v|\boldsymbol{x}_v) \neq p(\boldsymbol{y}_v|\boldsymbol{x}_v, \boldsymbol{x}_{v'})$ and $p(\boldsymbol{y}_v|\boldsymbol{x}_v) \neq p(\boldsymbol{y}_v|\boldsymbol{x}_v, \boldsymbol{y}_{v'})$ in general. Similarly, for node pairs, their labels are often conditionally-dependent, i.e., $p(\boldsymbol{y}_v, \boldsymbol{y}_{v'}|\boldsymbol{x}_v, \boldsymbol{x}_{v'}) = p(\boldsymbol{y}_v|\boldsymbol{x}_v, \boldsymbol{x}_{v'}, \boldsymbol{y}_{v'})p(\boldsymbol{y}_{v'}|\boldsymbol{x}_v, \boldsymbol{x}_{v'}) \neq p(\boldsymbol{y}_v|\boldsymbol{x}_v, \boldsymbol{x}_{v'})p(\boldsymbol{y}_{v'}|\boldsymbol{x}_v, \boldsymbol{x}_{v'})$. More generally, for all node labels, we have

$$p(\{\boldsymbol{y}_v\}_{v \in \mathcal{V}}|\boldsymbol{X}, \boldsymbol{A}) \neq \prod_{v \in \mathcal{V}} p(\boldsymbol{y}_v|\boldsymbol{X}, \boldsymbol{A}), \tag{1}$$

Ideally, we shall learn a GNN to maximize the conditional probability of all labeled nodes, i.e., $\max p(\{\boldsymbol{y}_v\}_{v \in \mathcal{V}_L}|\boldsymbol{X}, \boldsymbol{A})$. In practice, however, most existing methods formulate the node-level learning paradigm of the GNN as

$$\max_\theta \prod_{v \in \mathcal{V}_L} p(\boldsymbol{y}_v|\boldsymbol{X}, \boldsymbol{A}; \theta) \Leftrightarrow \min_\theta \sum_{v \in \mathcal{V}_L} \psi(g_v(\boldsymbol{X}, \boldsymbol{A}; \theta), \boldsymbol{y}_v). \tag{2}$$

Here, $g$ is a graph neural network whose parameters are denoted as $\theta$. Taking the adjacency matrix $\boldsymbol{A}$ and the node feature matrix $\boldsymbol{X}$ as input, the GNN $g$ predicts the node labels. $g_v(\boldsymbol{X}, \boldsymbol{A}; \theta)$ represents the estimation of the node $v$'s label achieved by the GNN, which is also denoted as $\hat{\boldsymbol{y}}_v$. Similarly, we denote $g_{\mathcal{V}}(\boldsymbol{X}, \boldsymbol{A}; \theta)$ as the estimated labels for the node set $\mathcal{V}$ in the following content. The loss function $\psi : \mathbb{R}^C \times \mathbb{R}^C \mapsto \mathbb{R}$ is defined in the node level. In node-level classification tasks, it is often implemented as the cross-entropy loss or the KL-divergence (i.e., $p(\boldsymbol{y}_v|\boldsymbol{X}, \boldsymbol{A}; \theta)$ is modeled by the softmax function). In node-level regression tasks, it is often implemented as the least-square loss (i.e., $p(\boldsymbol{y}_v|\boldsymbol{X}, \boldsymbol{A}; \theta)$ is assumed to be the Gaussian distribution).

**The loss in** (2) **assumes the node labels to be conditionally-independent with each other, which may be too strong in practice and inconsistent with the non-i.i.d. property of graph-structured data shown in** (1). To eliminate such inconsistency, we should treat node labels as a set rather than independent individuals, developing a set-level loss to penalize the discrepancy between the observed labels and their estimations globally, i.e.,

$$\min_\theta \text{Loss}(g_{\mathcal{V}_L}(\boldsymbol{X}, \boldsymbol{A}; \theta), \{\boldsymbol{y}_v\}_{v \in \mathcal{V}_L}). \tag{3}$$

In the following content, we will design such a loss with theoretical supports, based on the optimal transport on graphs.

## 3.2 Optimal Transport on Graphs

Suppose that we have two measures on a graph $G(\mathcal{V}, \mathcal{E})$, denoted as $\boldsymbol{\mu} \in [0, \infty)^{|\mathcal{V}|}$ and $\boldsymbol{\gamma} \in [0, \infty)^{|\mathcal{V}|}$, respectively. The element of each measure indicates the "mass" of a node. Assume the two measures to be balanced, i.e., $\langle \boldsymbol{\gamma} - \boldsymbol{\mu}, \mathbf{1}_{|\mathcal{V}|} \rangle = 0$, where $\langle \cdot, \cdot \rangle$ is the inner product operator. The optimal transport, or called the 1-Wasserstein distance [43], between them is defined as

$$W_1(\boldsymbol{\mu}, \boldsymbol{\gamma}) := \min_{T \in \Pi(\boldsymbol{\mu}, \boldsymbol{\gamma})} \langle \boldsymbol{D}, \boldsymbol{T} \rangle = \min_{T \in \Pi(\boldsymbol{\mu}, \boldsymbol{\gamma})} \sum_{v, v' \in \mathcal{V} \times \mathcal{V}} t_{vv'} d_{vv'}, \tag{4}$$

where $\boldsymbol{D} = [d_{vv'}] \in \mathbb{R}^{|\mathcal{V}| \times |\mathcal{V}|}$ represents the shortest path distance matrix, and $\Pi(\boldsymbol{\mu}, \boldsymbol{\gamma}) = \{\boldsymbol{T} \geq \boldsymbol{0} | \boldsymbol{T} \mathbf{1}_{|\mathcal{V}|} = \boldsymbol{\mu}, \boldsymbol{T}^\top \mathbf{1}_{|\mathcal{V}|} = \boldsymbol{\gamma}\}$ represents the set of all valid doubly stochastic matrices. Each $\boldsymbol{T} = [t_{vv'}] \in \Pi(\boldsymbol{\mu}, \boldsymbol{\gamma})$ is a transport plan matrix. The optimization problem in (4) corresponds to finding the optimal transport plan $\boldsymbol{T}^* = [t_{vv'}^*]$ to minimize the "cost" of changing $\boldsymbol{\mu}$ to $\boldsymbol{\gamma}$, in which the cost is measured as the sum of "mass" $t_{vv'}$ moved from node $v$ to node $v'$ times distance $d_{vv'}$.

### 3.2.1 Wasserstein Distance for Vectors on A Graph.
For the optimal transport problem defined on graphs, we can simplify the problem in (4) by leveraging the underlying graph structures. As shown in [12, 37], given a graph $G(\mathcal{V}, \mathcal{E})$, we can define a sparse matrix $\boldsymbol{S}_{\mathcal{V}} = [s_{ve}] \in \{0, \pm 1\}^{|\mathcal{V}| \times |\mathcal{E}|}$ to indicating the graph topology. For node $v$ and edge $e$, the corresponding element in $\boldsymbol{S}_{\mathcal{V}}$ is

$$s_{ve} = \begin{cases} 1 & \text{if } v \text{ is "head" of edge } e \\ -1 & \text{if } v \text{ is "tail" of edge } e \\ 0 & \text{otherwise.} \end{cases} \tag{5}$$

When $G$ is directed, the "head" and "tail" of each edge are predefined. When $G$ is undirected, we can randomly define each edge's "head" and "tail". Accordingly, the 1-Wasserstein distance in (4) can be equivalently formulated as a minimum-cost flow problem:

$$W_1(\boldsymbol{\mu}, \boldsymbol{\gamma}) = \min_{f \in \Omega(\boldsymbol{S}_{\mathcal{V}}, \boldsymbol{\mu}, \boldsymbol{\gamma})} \|\text{diag}(\boldsymbol{w})\boldsymbol{f}\|_1, \tag{6}$$

where $\text{diag}(\boldsymbol{w})$ is a diagonal matrix constructed by the edge weights of the graph. The vector $\boldsymbol{f} = [f_e] \in \Omega(\boldsymbol{S}_{\mathcal{V}}, \boldsymbol{\mu}, \boldsymbol{\gamma})$ is the flow indicating the mass passing through each edge. Accordingly, the cost corresponding to edge $e$ is represented as the distance $w_e$ times the mass $f_e$, and the sum of all the costs leads to the objective function in (6). The feasible domain of the flow vector is defined as

$$\Omega(\boldsymbol{S}_{\mathcal{V}}, \boldsymbol{\mu}, \boldsymbol{\gamma}) = \mathcal{U}^{|\mathcal{E}|} \cap \{\boldsymbol{f} \mid \boldsymbol{S}_{\mathcal{V}} \boldsymbol{f} = \boldsymbol{\gamma} - \boldsymbol{\mu}\},$$
$$\text{where } \mathcal{U} = \begin{cases} [0, \infty), & \text{Directed } G, \\ \mathbb{R}, & \text{Undirected } G. \end{cases} \tag{7}$$

The constraint $\boldsymbol{S}_{\mathcal{V}} \boldsymbol{f} = \boldsymbol{\gamma} - \boldsymbol{\mu}$ ensures that the flow on all the edges leads to the change from $\boldsymbol{\mu}$ to $\boldsymbol{\gamma}$. The flow in the directed graph is nonnegative, which can only pass through each edge from "head" to "tail". On the contrary, the undirected graph allows the mass to transport from "tail" to "head". By solving (6), we find the optimal flow $\boldsymbol{f}^*$ (or the so-called optimal transport on edges) that minimizes the overall cost $\|\text{diag}(\boldsymbol{w})\boldsymbol{f}\|_1$.

### 3.2.2 Partial Wasserstein Distance for Vectors on a Graph.
When only a subset of nodes (i.e., $\mathcal{V}_L \subset \mathcal{V}$) have observable signals, we can define a *partial Wasserstein distance* on the graph by preserving the constraints relevant to $\mathcal{V}_L$: for $\boldsymbol{\mu}, \boldsymbol{\gamma} \in \mathbb{R}^{|\mathcal{V}_L|}$, we have

$$W_1^{(P)}(\boldsymbol{\mu}, \boldsymbol{\gamma}) = \min_{f \in \Omega(\boldsymbol{S}_{\mathcal{V}_L}, \boldsymbol{\mu}, \boldsymbol{\gamma})} \|\text{diag}(\boldsymbol{w})\boldsymbol{f}\|_1, \tag{8}$$

where $\boldsymbol{S}_{\mathcal{V}_L} \in \{0, \pm 1\}^{|\mathcal{V}_L| \times |\mathcal{E}|}$ is a submatrix of $\boldsymbol{S}_{\mathcal{V}}$, storing the rows corresponding to the nodes with observable signals.

Different from the classic Wasserstein distance, the Wasserstein distance defined on graphs is not limited for nonnegative vectors because the minimum-cost flow formulation in (6) is *shift-invariant*, i.e., $\forall \boldsymbol{\delta} \in \mathbb{R}^{|\mathcal{V}|}, W_1(\boldsymbol{\mu}, \boldsymbol{\gamma}) = W_1(\boldsymbol{\mu} - \boldsymbol{\delta}, \boldsymbol{\gamma} - \boldsymbol{\delta})$. The partial Wasserstein distance in (8) can be viewed as a generalization of (6), and thus it holds the shift-invariance as well. Moreover, given an undirected graph, we can prove that both the $W_1$ in (6) and the $W_1^{(P)}$ in (8) can be valid metrics in the spaces determined by the topology of an undirected graph (See Appendix A for a complete proof).

THEOREM 1 (THE VALIDNESS AS A METRIC). *Given an undirected graph* $G(\mathcal{V}, \mathcal{E})$, *with edge weights* $\boldsymbol{w} \in [0, \infty)^{|\mathcal{E}|}$ *and a matrix* $\boldsymbol{S}_{\mathcal{V}} \in \{0, \pm 1\}^{|\mathcal{V}| \times \mathcal{E}}$ *defined in* (5), *the* $W_1$ *in* (6) *is a metric in Range*$(\boldsymbol{S}_{\mathcal{V}})$, *and the* $W_1^{(P)}$ *in* (8) *is a metric in Range*$(\boldsymbol{S}_{\mathcal{V}_L})$, $\forall \mathcal{V}_L \in \mathcal{V}$.

The flow vector $f$ in (8) has fewer constraints than that in (6), so the relation between $W_1$ and $W_1^{(P)}$ obeys the following theorem.

THEOREM 2 (THE MONOTONICITY). *Given an undirected graph $G(\mathcal{V}, \mathcal{E})$, with edge weights $w \in [0, \infty)^{|\mathcal{E}|}$ and a matrix $S_{\mathcal{V}} \in \{0, \pm1\}^{|\mathcal{V}| \times \mathcal{E}}$ defined in (5), we have*

$$W_1(\mu, \gamma) \geq W_1^{(P)}(\mu_{\mathcal{V}'}, \gamma_{\mathcal{V}'}) \geq W_1^{(P)}(\mu_{\mathcal{V}''}, \gamma_{\mathcal{V}''}) \geq 0,$$
$$\forall \mu, \gamma \in Range(S_{\mathcal{V}}), \ \mathcal{V}'' \subset \mathcal{V}' \subset \mathcal{V},$$
(9)

*where $\mu_{\mathcal{V}'}$ denotes the subvector of $\mu$ corresponding to the set $\mathcal{V}'$.*

## 3.3 Learning GNNs with Quasi-Wasserstein Loss

Inspired by the optimal transport on graphs, we propose a Quasi-Wasserstein loss for learning GNNs. Given a partially-labeled graph $G$, we formulate observed labels as a matrix $Y_{\mathcal{V}_L} = [y_{\mathcal{V}_L}^{(c)}] \in \mathbb{R}^{|\mathcal{V}_L| \times C}$, where $y_{\mathcal{V}_L}^{(c)} \in \mathbb{R}^{|\mathcal{V}_L|}$ represents the labels in $c$-th dimension. The estimated labels generated by a GNN $g$ are represented as $\widehat{Y}_{\mathcal{V}_L} = [\hat{y}_{\mathcal{V}_L}^{(c)}] := g_{\mathcal{V}_L}(X, A; \theta)$. Our QW loss is defined as

$$QW(\widehat{Y}_{\mathcal{V}_L}, Y_{\mathcal{V}_L}) = \sum_{c=1}^{C} W_1^{(P)}(\hat{y}_{\mathcal{V}_L}^{(c)}, y_{\mathcal{V}_L}^{(c)})$$
$$= \sum_{c=1}^{C} \min_{f^{(c)} \in \Omega(S_{\mathcal{V}_L}, \hat{y}_{\mathcal{V}_L}^{(c)}, y_{\mathcal{V}_L}^{(c)})} \|\text{diag}(w) f^{(c)}\|_1$$
$$= \min_{F \in \Omega_C(S_{\mathcal{V}_L}, g_{\mathcal{V}_L}(X, A; \theta), Y_{\mathcal{V}_L})} \|\text{diag}(w) F\|_1,$$
(10)

where $F = [f^{(c)}]$ is the flow matrix, and its feasible domain $\Omega_C = \mathcal{U}^{|\mathcal{E}| \times C} \cap \{F | S_{\mathcal{V}_L} F = Y_{\mathcal{V}_L} - g_{\mathcal{V}_L}(X, A; \theta)\}$. $W_1^{(P)}(\hat{y}_{\mathcal{V}_L}^{(c)}, y_{\mathcal{V}_L}^{(c)})$ is the partial Wasserstein distance between the observed and estimated labels in the $c$-th dimension. The QW loss is the summation of the $C$ partial Wasserstein distances, and it can be rewritten as the summation of $C$ minimum-cost flow problems. As shown in the third row of (10), these $C$ problems can be formulated as a single optimization problem, and the variables of the problems are aggregated as the flow matrix $F$. Based on Theorem 1, our QW loss is a metric for the matrices whose columns are in $Range(S_{\mathcal{V}_L})$.

The optimal flow matrix $F^* = [f^{(c)*}]$ is called **optimal label transport** in this study, in which the column $f^{(c)*}$ indicating the optimal transport on graph edges between the observed labels and their estimations in the $c$-th dimension. Note that, we call the proposed loss "Quasi-Wasserstein" because $i$) it is not equal to the classic 1-Wasserstein distance between the label sets $\widehat{Y}_{\mathcal{V}_L}$ and $Y_{\mathcal{V}_L}$ [34], and $ii$) the optimal label transport $F^*$ cannot be reformulated as the optimal transport $T^*$ obtained by (4). When $F^* = 0$, we have $QW(\widehat{Y}_{\mathcal{V}_L}, Y_{\mathcal{V}_L}) = 0$ and accordingly, $\widehat{Y}_{\mathcal{V}_L} = Y_{\mathcal{V}_L}$, which means that the GNN $g$ perfectly estimates the observed labels. Therefore, the QW loss provides a new alternative for the training loss of GNN. Different from the traditional loss function in (2), the QW loss treats observed node labels and their estimations as two sets and measures their discrepancy accordingly, which provides an effective implementation of the loss in (3).

Applying the QW loss to learn a GNN $g$ results in the following constrained optimization problem:

$$\min_\theta QW(g_{\mathcal{V}_L}(X, A; \theta), Y_{\mathcal{V}_L})$$
$$= \min_\theta \min_{F \in \Omega_C(S_{\mathcal{V}_L}, g_{\mathcal{V}_L}(X, A; \theta), Y_{\mathcal{V}_L})} \|\text{diag}(w) F\|_1.$$
(11)

To solve it effectively, we consider the following two algorithms.

---

**Algorithm 1** Learning a GNN by solving (12)

**Require:** A graph $G$, its adjacency matrix $A$ (edge weights $w$), node features $X$, and observed labels $Y_{\mathcal{V}_L}$.
1: Initialize $\theta^{(0)}$ and $F^{(0)}$ randomly.
2: **while** Not converge **do**
3:      Compute Loss $= \|\text{diag}(w) F\|_1 + \lambda B_\phi(g_{\mathcal{V}_L}(X, A; \theta) + S_{\mathcal{V}_L} F, Y_{\mathcal{V}_L})$
4:      Update $\{F, \theta\}$ by Adam [25].
5:      **if** $G$ is a directed graph **then** $F \leftarrow \text{Proj}_{\geq 0}(F)$ **end if**
6: **end while**
7: **return** Optimal label transport $F^*$ and model parameters $\theta^*$.

---

*3.3.1 Bregman Divergence-based Approximate Solver.* By relaxing the constraint $\{F | S_{\mathcal{V}_L} F = Y_{\mathcal{V}_L} - g_{\mathcal{V}_L}(X, A; \theta)\}$ to a Bregman divergence-based regularizer [3], we can reformulate (11) to the following problem:

$$\min_{\theta, \ F \in \mathcal{U}^{|\mathcal{E}| \times C}} \|\text{diag}(w) F\|_1 + \lambda \underbrace{B_\phi(g_{\mathcal{V}_L}(X, A; \theta) + S_{\mathcal{V}_L} F, \ Y_{\mathcal{V}_L})}_{\sum_{v \in \mathcal{V}_L} \psi(g_v(X, A; \theta) + S_v F, \ y_v)}.$$
(12)

Here, $B_\phi(x, y) = \phi(x) - \phi(y) - \langle \nabla \phi(y), x - y \rangle$ is the Bregman divergence defined based on the strictly convex function $\phi$, and the hyperparameter $\lambda > 0$ controls its significance. In node classification tasks, we can set $\phi$ as an entropic function, and the Bregman divergence becomes the KL-divergence. In node regression tasks, we can set $\phi$ as a least-square loss, and the Bregman divergence becomes the least-square loss. Therefore, as shown in (12), the Bregman divergence $B_\phi$ can be implemented based on commonly-used loss functions, e.g., the $\psi$ in (2).

Algorithm 1 shows the algorithmic scheme in details. For undirected graphs, the $\mathcal{U}$ in (12) is $\mathbb{R}$, and accordingly, (12) becomes a unconstrained optimization problem. We can solve it efficiently by gradient descent. For directed graphs, we just need to add a projection step when updating the flow matrix $F$, leading to the projected gradient descent algorithm.

*3.3.2 Bregman ADMM-based Exact Solver.* Algorithm 1 solves (11) approximately — because of relaxing the strict equality constraint to a regularizer, the solution of (12) often cannot satisfy the original equality constraint. To solve (11) exactly, we further develop a learning method based on the Bregman ADMM (Bremgan Alternating Direction Method of Multipliers) algorithm [45]. In particular, we can rewrite (11) in the following augmented Lagrangian form:

$$\min_{\theta, \ F \in \mathcal{U}^{|\mathcal{E}| \times C}, \ Z \in \mathbb{R}^{|\mathcal{V}_L| \times C}} \|\text{diag}(w) F\|_1$$
$$+ \langle Z, \ g_{\mathcal{V}_L}(X, A; \theta) + S_{\mathcal{V}_L} F - Y_{\mathcal{V}_L} \rangle$$
$$+ \lambda B_\phi(g_{\mathcal{V}_L}(X, A; \theta) + S_{\mathcal{V}_L} F, \ Y_{\mathcal{V}_L}),$$
(13)

where $Z \in \mathbb{R}^{|\mathcal{V}_L| \times C}$ is the dual variable, the second term in (13) is the Lagrangian term corresponding to the equality constraint, and the third term in (13) is the augmented term implemented as the Bregman divergence.

Denote the objective function in (13) as $L(\theta, F, Z)$. In the Bregman ADMM framework, we can optimize $\theta$, $F$, and $Z$ iteratively through alternating optimization. In the $k$-th iteration, we update

---

**Algorithm 2** Learning a GNN by solving (13)

**Require:** A graph $G$, its adjacency matrix $A$ (edge weights $w$), node features $X$, observed labels $Y_{\mathcal{V}_L}$, the number of inner iterations $J$.
1: Initialize $\theta^{(0)}$ and $F^{(0)}$ randomly and set $Z^{(0)} = \mathbf{0}_{|\mathcal{V}_L| \times C}$.
2: **while** Not converge **do**
3:     Solve (14) by Adam [25] with $J$ steps and obtain $\theta^{(k+1)}$.
4:     Solve (15) by Adam [25] with $J$ steps and obtain $F^{(k+1)}$.
5:     **if** $G$ is a directed graph **then** $F^{(k+1)} \leftarrow \text{Proj}_{\geq 0}(F^{(k+1)})$ **end if**
6:     Obtain $Z^{(k+1)}$ by (16).
7: **end while**
8: **return** Optimal label transport $F^*$ and model parameters $\theta^*$.

---

the three variables via solving the following three subproblems:

$$\theta^{(k+1)} = \arg\min_\theta L(\theta, F^{(k)}, Z^{(k)})$$
$$= \arg\min_\theta \langle Z^{(k)}, \; g_{\mathcal{V}_L}(X, A; \theta) \rangle \tag{14}$$
$$+ \lambda B_\phi(g_{\mathcal{V}_L}(X, A; \theta) + S_{\mathcal{V}_L} F^{(k)}, \; Y_{\mathcal{V}_L}).$$

$$F^{(k+1)} = \arg\min_{F \in \mathcal{U}^{|\mathcal{E}| \times C}} L(\theta^{(k+1)}, F, Z^{(k)})$$
$$= \arg\min_{F \in \mathcal{U}^{|\mathcal{E}| \times C}} \langle Z^{(k)}, \; S_{\mathcal{V}_L} F \rangle \tag{15}$$
$$+ \lambda B_\phi(g_{\mathcal{V}_L}(X, A; \theta^{(k+1)}) + S_{\mathcal{V}_L} F, \; Y_{\mathcal{V}_L}).$$

$$Z^{(k+1)} = Z^{(k)} + \lambda(g_{\mathcal{V}_L}(X, A; \theta^{(k+1)}) + S_{\mathcal{V}_L} F^{(k+1)} - Y_{\mathcal{V}_L}). \tag{16}$$

We can find that (14) is a unconstrained optimization problem, so we can update the model parameter $\theta$ by gradient descent. Similarly, we can solve (15) and update the flow matrix $F$ by gradient descent or projected gradient descent, depending on whether the graph is undirected or not. Finally, the update of the dual variable $Z$ can be achieved in a closed form, as shown in (16). The Bregman ADMM algorithm solves the original problem in (12) rather than a relaxed version. In theory, with the increase of iterations, we can obtain the optimal variables that satisfy the equality constraint. Algorithm 2 shows the scheme of the Bregman ADMM-based solver.

*3.3.3 Optional Edge Weight Prediction.* Some GNNs, e.g., GCN-LPA [46] and GAT [42], model the adjacency matrix of graph as learnable parameters. Inspired by these models, we can optionally introduce an edge weight predictor and parameterize the adjacency matrix based on the flow matrix, as shown in Figure 1. In particular, given $F$, we can apply a multi-layer perceptron (MLP) to embed it to an edge weight vector and then obtain a weighted adjacency matrix. Accordingly, the learning problem becomes

$$\min_{\theta, \xi} \; \min_{F \in \Omega_C(S_{\mathcal{V}_L}, g_{\mathcal{V}_L}(X, A(F; \xi); \theta), Y_{\mathcal{V}_L})} \|\text{diag}(w) F\|_1, \tag{17}$$

where $A(F; \xi)$ represents the adjacency matrix determined by the label transportation $F$, and $\xi$ represents the parameters of the MLP. Taking the learning of $\xi$ into account, we can modify the above two solvers slightly and make them applicable for solving (17).

## 3.4 Connections to Traditional Methods

As discussed in Section 3.3.1 and shown in (12), the Bregman divergence $B_\phi$ can be implemented as the commonly-used loss function $\psi$ in (2) (e.g., the KL divergence or the least-square loss). Table 1 compares the typical setting of traditional learning methods and that of our QW loss-based method in different node-level tasks.

**Table 1: Comparison between traditional methods and ours**

| Method | Setting | Node Classification | Node Regression |
|---|---|---|---|
| Apply the loss in (2) | $\psi$ | Cross-entropy or KL | Least-square |
| | Predicted $y_v$ | $g_v(X, A; \theta), \forall v \in \mathcal{V} \setminus \mathcal{V}_L$ | |
| Apply the QW loss | $\phi$ | Entropy | $\frac{1}{2}\|\cdot\|_2^2$ |
| | $B_\phi(= \psi)$ | KL | Least-square |
| | Predicted $y_v$ | $g_v(X, A; \theta) + S_v F, \forall v \in \mathcal{V} \setminus \mathcal{V}_L$ | |

Essentially, the traditional learning method in (2) can be viewed as a special case of our QW loss-based learning method. In particular, when setting $F = \mathbf{0}_{|\mathcal{E}| \times C}$ and $B_\phi = \psi$, the objective function in (12) degrades to the objective function in (2), which treats each node independently. Similarly, when further setting the dual variable $Z = \mathbf{0}_{|\mathcal{V}_L| \times C}$, the objective function in (13) degrades to the objective function in (2) as well. In theory, we demonstrate that our QW loss-based learning method can fit training data better than the traditional method does, as shown in the following theorem.

THEOREM 3. *Let* $\{\theta^\star, F^\star, Z^\star\}$ *be the global optimal solution of* (13), $\{\theta^\dagger, F^\dagger\}$ *be the global optimal solution of* (12), *and* $\theta^\ddagger$ *be the global optimal solution of* $\min_\theta B_\phi(g_{\mathcal{V}_L}(X, A; \theta), Y_{\mathcal{V}_L})$, *we have*

$$B_\phi(g_{\mathcal{V}_L}(X, A; \theta^\star) + S_{\mathcal{V}_L} F^\star, \; Y_{\mathcal{V}_L})$$
$$\leq B_\phi(g_{\mathcal{V}_L}(X, A; \theta^\dagger) + S_{\mathcal{V}_L} F^\dagger, \; Y_{\mathcal{V}_L}) \leq B_\phi(g_{\mathcal{V}_L}(X, A; \theta^\ddagger), \; Y_{\mathcal{V}_L})$$

PROOF. The proof is straightforward — $\{\theta^\ddagger, \mathbf{0}_{|\mathcal{E}| \times C}\}$ is a feasible solution of (12), so the corresponding objective is equal to or larger than that obtained by $\{\theta^\dagger, F^\dagger\}$. Similarly, $\{\theta^\dagger, F^\dagger, \mathbf{0}_{|\mathcal{V}_L| \times C}\}$ is a feasible solution of (13), so the corresponding objective is equal to or larger than that obtained by $\{\theta^\star, F^\star, Z^\star\}$. □

**Remark.** It should be noted that although the Bregman ADMM-based solver can fit training data better in theory, in the cases with distribution shifting or out-of-distribution issues, it has a higher risk of over-fitting. Therefore, in practice, we can select one of the above two solvers to optimize the GW loss, depending on their performance. In the following experimental section, we will further compare these two solvers in details.

## 3.5 A New Transductive Prediction Paradigm

As shown in Table 1, given the learned model $\theta^*$ and the optimal label transport $F^*$, we predict node labels in a new transductive prediction paradigm. For $v \in \mathcal{V} \setminus \mathcal{V}_L$, we predict its label as

$$\tilde{y}_v := g_v(X, A; \theta^*) + S_v F^*, \tag{18}$$

which combines the estimated label from the learned GNN and the residual component from the optimal label transport.

Note that some attempts have been made to incorporate label propagation algorithms (LPAs) [59] into GNNs, e.g., the GCN-LPA in [46] and the FDiff-Scale in [23]. The PTA in [9] demonstrates that learning a decoupled GNN is equivalent to implementing a label propagation algorithm. These methods leverage LPAs to regularize the learning of GNNs. However, in the prediction phase, they abandon the training labels and rely only on the GNNs to predict node labels, as shown in Table 1. Unlike these methods, our QW loss-based method achieves a new kind of label propagation with

**Table 2: Basic information of the graphs and the comparisons on node classification accuracy (%).**

| Model | Method | Homophilic graphs | | | | | Heterophilic graphs | | | | | Overall Improve |
|---|---|---|---|---|---|---|---|---|---|---|---|---|
| | | Cora | Citeseer | Pubmed | Computers | Photo | Squirrel | Chameleon | Actor | Texas | Cornell | |
| #Nodes ($|\mathcal{V}|$) | | 2,708 | 3,327 | 19,717 | 13,752 | 5,201 | 7,650 | 2,277 | 7,600 | 183 | 183 | |
| #Features ($D$) | | 1,433 | 3,703 | 500 | 767 | 754 | 2,089 | 2,325 | 932 | 1,703 | 1,703 | |
| #Edges ($|\mathcal{E}|$) | | 5,278 | 4,552 | 44,324 | 245,861 | 119,081 | 198,358 | 31,371 | 26,659 | 279 | 277 | |
| Intra-edge rate | | 81.0% | 73.6% | 80.2% | 77.7% | 82.7% | 22.2% | 23.0% | 21.8% | 6.1% | 12.3% | |
| #Classes ($C$) | | 7 | 6 | 5 | 10 | 8 | 5 | 5 | 5 | 5 | 5 | |
| GCN | (2) | $87.44_{\pm0.96}$ | $79.98_{\pm0.84}$ | $86.93_{\pm0.29}$ | $88.42_{\pm0.45}$ | $93.24_{\pm0.43}$ | $46.55_{\pm1.15}$ | $63.57_{\pm1.16}$ | $34.00_{\pm1.28}$ | $77.21_{\pm3.28}$ | $61.91_{\pm5.11}$ | — |
| | (2)+LPA | $86.34_{\pm1.45}$ | $78.51_{\pm1.22}$ | $84.72_{\pm0.70}$ | $82.48_{\pm0.69}$ | $88.10_{\pm1.31}$ | $44.81_{\pm1.81}$ | $60.90_{\pm1.63}$ | $32.43_{\pm1.59}$ | $78.69_{\pm6.47}$ | $68.72_{\pm5.95}$ | -1.36 |
| | QW | $\mathbf{87.88_{\pm0.79}}$ | $\mathbf{81.36_{\pm0.41}}$ | $\mathbf{87.89_{\pm0.40}}$ | $\mathbf{89.20_{\pm0.41}}$ | $\mathbf{93.81_{\pm0.36}}$ | $\mathbf{52.62_{\pm0.49}}$ | $\mathbf{68.10_{\pm1.01}}$ | $\mathbf{38.09_{\pm0.50}}$ | $\mathbf{84.10_{\pm2.95}}$ | $\mathbf{84.26_{\pm2.98}}$ | +4.81 |
| GAT | (2) | $\mathbf{89.20_{\pm0.79}}$ | $\mathbf{80.75_{\pm0.78}}$ | $87.42_{\pm0.33}$ | $90.08_{\pm0.36}$ | $94.38_{\pm0.25}$ | $48.20_{\pm1.67}$ | $64.31_{\pm2.01}$ | $\mathbf{35.68_{\pm0.60}}$ | $80.00_{\pm3.11}$ | $68.09_{\pm2.13}$ | — |
| | QW | $89.11_{\pm0.66}$ | $80.19_{\pm0.64}$ | $\mathbf{88.38_{\pm0.23}}$ | $\mathbf{90.41_{\pm0.28}}$ | $\mathbf{94.65_{\pm0.24}}$ | $\mathbf{55.03_{\pm1.35}}$ | $\mathbf{67.35_{\pm1.42}}$ | $33.86_{\pm2.13}$ | $\mathbf{80.33_{\pm1.80}}$ | $\mathbf{70.21_{\pm2.13}}$ | +1.14 |
| GIN | (2) | $86.22_{\pm0.95}$ | $\mathbf{76.18_{\pm0.78}}$ | $\mathbf{87.87_{\pm0.23}}$ | $80.87_{\pm1.43}$ | $89.83_{\pm0.72}$ | $39.11_{\pm2.23}$ | $64.29_{\pm1.51}$ | $\mathbf{32.37_{\pm1.56}}$ | $72.79_{\pm4.92}$ | $62.55_{\pm4.80}$ | — |
| | QW | $\mathbf{86.24_{\pm0.90}}$ | $76.13_{\pm1.09}$ | $87.53_{\pm0.34}$ | $\mathbf{89.28_{\pm0.45}}$ | $\mathbf{92.60_{\pm0.44}}$ | $\mathbf{65.29_{\pm0.68}}$ | $\mathbf{73.26_{\pm1.12}}$ | $32.32_{\pm1.93}$ | $\mathbf{77.54_{\pm2.60}}$ | $\mathbf{64.04_{\pm3.62}}$ | +5.22 |
| GraphSAGE | (2) | $\mathbf{88.24_{\pm0.95}}$ | $79.81_{\pm0.80}$ | $88.14_{\pm0.25}$ | $89.71_{\pm0.38}$ | $95.08_{\pm0.26}$ | $43.79_{\pm0.59}$ | $63.26_{\pm1.09}$ | $\mathbf{38.99_{\pm0.85}}$ | $90.00_{\pm2.30}$ | $84.26_{\pm2.98}$ | — |
| | QW | $87.59_{\pm0.77}$ | $\mathbf{80.52_{\pm0.68}}$ | $\mathbf{88.61_{\pm0.32}}$ | $\mathbf{90.17_{\pm0.24}}$ | $\mathbf{95.25_{\pm0.25}}$ | $\mathbf{54.37_{\pm0.89}}$ | $\mathbf{68.32_{\pm0.68}}$ | $37.82_{\pm0.45}$ | $\mathbf{90.33_{\pm1.97}}$ | $\mathbf{86.38_{\pm2.13}}$ | +1.18 |
| APPNP | (2) | $88.14_{\pm0.73}$ | $80.47_{\pm0.74}$ | $88.12_{\pm0.31}$ | $85.32_{\pm0.37}$ | $88.51_{\pm0.31}$ | $36.15_{\pm0.75}$ | $52.93_{\pm1.71}$ | $40.46_{\pm0.64}$ | $\mathbf{91.31_{\pm1.97}}$ | $87.66_{\pm2.13}$ | — |
| | QW | $\mathbf{88.74_{\pm0.84}}$ | $\mathbf{80.94_{\pm0.61}}$ | $\mathbf{89.48_{\pm0.28}}$ | $\mathbf{86.95_{\pm0.82}}$ | $\mathbf{94.43_{\pm0.24}}$ | $\mathbf{38.73_{\pm1.06}}$ | $\mathbf{53.76_{\pm1.25}}$ | $\mathbf{40.78_{\pm0.74}}$ | $91.48_{\pm2.30}$ | $\mathbf{87.87_{\pm2.34}}$ | +1.41 |
| BernNet | (2) | $88.28_{\pm1.00}$ | $79.81_{\pm0.79}$ | $88.87_{\pm0.38}$ | $87.61_{\pm0.46}$ | $93.68_{\pm0.28}$ | $51.15_{\pm1.09}$ | $67.96_{\pm1.05}$ | $40.72_{\pm0.80}$ | $93.28_{\pm1.48}$ | $90.21_{\pm2.35}$ | — |
| | QW | $\mathbf{89.03_{\pm0.76}}$ | $\mathbf{81.35_{\pm0.71}}$ | $\mathbf{89.03_{\pm0.38}}$ | $\mathbf{89.58_{\pm0.47}}$ | $\mathbf{94.55_{\pm0.39}}$ | $\mathbf{55.22_{\pm0.64}}$ | $71.66_{\pm1.18}$ | $\mathbf{40.91_{\pm0.71}}$ | $\mathbf{93.44_{\pm1.80}}$ | $\mathbf{90.85_{\pm2.34}}$ | +1.41 |
| ChebNetII | (2) | $88.26_{\pm0.89}$ | $\mathbf{80.00_{\pm0.74}}$ | $88.57_{\pm0.36}$ | $86.58_{\pm0.71}$ | $93.50_{\pm0.34}$ | $57.78_{\pm0.84}$ | $71.71_{\pm1.40}$ | $40.70_{\pm0.77}$ | $92.79_{\pm1.48}$ | $\mathbf{88.94_{\pm2.78}}$ | — |
| | QW | $\mathbf{88.54_{\pm0.76}}$ | $79.47_{\pm0.70}$ | $\mathbf{89.47_{\pm0.36}}$ | $\mathbf{90.43_{\pm0.22}}$ | $\mathbf{94.84_{\pm0.37}}$ | $\mathbf{60.55_{\pm0.64}}$ | $\mathbf{74.05_{\pm0.68}}$ | $\mathbf{41.37_{\pm0.67}}$ | $\mathbf{93.93_{\pm0.98}}$ | $87.23_{\pm3.62}$ | +1.11 |

the help of computational optimal transport, saving the training label information in the optimal label transport and applying it explicitly in the prediction phase.

## 4 EXPERIMENTS

To demonstrate the effectiveness of our QW loss-based learning method, we apply it to learn GNNs with various architectures and test the learned GNNs in different node-level prediction tasks. We compare our learning method with the traditional one in (2) on their model performance and computational efficiency. For our method, we conduct a series of analytic experiments to show its robustness to hyperparameter settings and label insufficiency. All the experiments are conducted on a machine with three NVIDIA A40 GPUs, and the code is implemented based on PyTorch.

### 4.1 Implementation Details

*4.1.1 Datasets.* The datasets we considered consist of five homophilic graphs (i.e., **Cora**, **Citeseer**, **Pubmed** [39, 56], **Computers**, and **Photo** [31]) and five heterophilic graphs (i.e., **Chameleon**, **Squirrel** [36], **Actor**, **Texas**, and **Cornell** [33]), respectively. Following the work in [46], we categorize the graphs according to the percentage of the edges connecting the nodes of the same class (i.e., the intra-edge rate). The basic information of these graphs is shown in Table 2. Additionally, a large **arXiv-year** graph [30] is applied to demonstrate the efficiency of our method. The adjacency matrix of each graph is binary, so the edge weights $\boldsymbol{w} = \mathbf{1}_{|\mathcal{E}|}$.

*4.1.2 GNN Architectures.* In the following experiments, the models we considered include *i*) the representative spatial GNNs, i.e., **GCN** [26], **GAT** [42], **GIN** [55], **GraphSAGE** [20], and **GCN-LPA** [46] that combines the GCN with the label propagation algorithm; and *ii*) state-of-the-art spectral GNNs, i.e., **APPNP** [17],

**BernNet** [22], and **ChebNetII** [21]. For a fair comparison, we set the architectures of the GNNs based on the code provided by [21] and configure the algorithmic hyperparameters by grid search. More details of the hyperparameter settings are in Appendix.

*4.1.3 Learning Tasks and Evaluation Measurements.* For each graph, their nodes belong to different classes. Therefore, we first learn different GNNs to solve the node-level classification tasks defined on the above graphs. By default, the split ratio of each graph's nodes is 60% for training, 20% for validation, and 20% for testing, respectively. The GNNs are learned by *i*) the traditional learning method in (2)[1] and *ii*) minimizing the proposed QW loss in (11), respectively. When implementing the QW loss, we apply either Algorithm 1 or 2, depending on their performance. Additionally, to demonstrate the usefulness of our QW loss in node-level regression tasks, we treat the node labels as one-hot vectors and fit them by minimizing the mean squared error (MSE), in which the $\psi$ in (2) and the corresponding Bregman divergence $B_\phi$ are set to be the least-square loss. For each method, we perform 10 runs with different seeds and record the learning results' mean and standard deviation.

### 4.2 Numerical Comparison and Visualization

*4.2.1 Node Classification and Regression.* Table 2 shows the node classification results on the ten graphs,[2] whose last column records the overall improvements caused by our QW loss compared to other learning methods. The experimental results demonstrate the usefulness of our QW loss-based learning method — for each model, applying our QW loss helps to improve learning results in most situations and leads to consistent overall improvements. In particular,

---

[1]For GCN-LPA [46], it learns a GCN model by imposing a label propagation-based regularizer on (2) and adjusting edge weights by the propagation result.
[2]In Table 2, we bold the best learning result for each graph. Learning GCN by "(2)+LPA" means implementing GCN-LPA [46].

**Table 3: Comparisons on node regression error (MSE).**

| Model | Method | Homophilic graphs | | Heterophilic graphs | |
|---|---|---|---|---|---|
| | | Computers | Photo | Actor | Cornell |
| GIN | (2) | $0.0605_{\pm0.0018}$ | $0.0459_{\pm0.0044}$ | $0.1570_{\pm0.0014}$ | $0.1609_{\pm0.0359}$ |
| | QW | $\mathbf{0.0244_{\pm0.0028}}$ | $\mathbf{0.0203_{\pm0.0012}}$ | $\mathbf{0.1564_{\pm0.0012}}$ | $\mathbf{0.1524_{\pm0.0043}}$ |
| BernNet | (2) | $0.0871_{\pm0.0002}$ | $0.0488_{\pm0.0009}$ | $\mathbf{0.1661_{\pm0.0020}}$ | $0.0989_{\pm0.0076}$ |
| | QW | $\mathbf{0.0364_{\pm0.0038}}$ | $\mathbf{0.0297_{\pm0.0014}}$ | $0.1671_{\pm0.0008}$ | $\mathbf{0.0753_{\pm0.0024}}$ |

for the state-of-the-art spectral GNNs like BernNet [22] and ChebNetII [21], learning with our QW loss can improve their overall performance on both homophilic and heterophilic graphs consistently, resulting in the best performance in this experiment. For the simple GCN model [26], learning with our QW loss improves its performance significantly and reduces the gap between its classification accuracy and that of the state-of-the-art models [17, 21, 22], especially heterophilic graphs. Note that, when learning GCN, our QW loss works better than the traditional method regularized by the label propagation (i.e., GCN-LPA [46]) because our learning method can leverage the training label information in both learning and prediction phases. GCN-LPA improves GCN when learning on heterophilic graphs, but surprisingly, leads to performance degradation on homophilic graphs. Additionally, we also fit the one-hot labels by minimizing the MSE. As shown in Table 3, minimizing the QW loss leads to lower MSE results, which demonstrates the usefulness of the QW loss in node-level regression tasks.

*4.2.2 Computational Efficiency and Scalability.* In theory, the computational complexity of our QW loss is linear with the number of edges. When implementing the loss as (13) and solving ti by Algorithm 2, its complexity is also linear with the number of inner iterations $J$. Figure 2 shows the runtime comparisons for the QW loss-based learning methods and the traditional method on two datasets. We can find that minimizing the QW loss by Algorithm 1 or Algorithm 2 with $J = 1$ merely increases the training time slightly compared to the traditional method. Empirically, setting $J \leq 5$ can leads to promising learning results, as shown in Tables 2 and 3. In other words, the computational cost of applying the QW loss is tolerable considering the significant performance improvements it achieved. Additionally, we apply our QW loss to large-scale graphs and test its scalability. As shown in Table 4, we implement the QW loss based on Algorithm 1 (i.e., solving (12)) and apply it to the node classification task in the large-scale arXiv-year graph. The result shows that our QW loss is applicable to the graphs with millions of edges on a single GPU and improves the model performance.

*4.2.3 Distribution of Optimal Label Transport.* Figure 3 visualizes the histograms of the optimal label transport $F^*$ learned for two representative GNNs (i.e., GCN [26] and ChebNetII [21]) on four graphs (i.e., the homophilic graphs "Computers" and "Photo" and the heterophilic graphs "Squirrel" and "Chameleon"). We can find that when learning on homophilic graphs, the elements of the optimal label transport obey the zero-mean Laplacian distribution. It is reasonable from the perspective of optimization — the term $\|\text{diag}(\boldsymbol{w})\boldsymbol{F}\|_1$ can be explained as a Laplacian prior imposed on $F$'s element. Additionally, we can find that the distribution corresponding to GCN has larger variance than that corresponding to

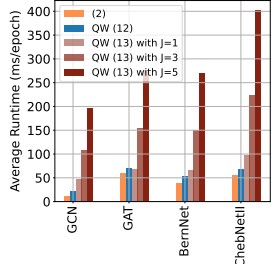

| Graph | arXiv-year |
|---|---|
| #Nodes ($|\mathcal{V}|$) | 169,343 |
| #Features ($D$) | 128 |
| #Edges ($|\mathcal{E}|$) | 1,166,243 |
| Intra-edge rate | 22.0% |
| #Classes ($C$) | 5 |
| ChebNetII   (2) | $48.18_{\pm0.18}$ |
| QW | $\mathbf{48.30_{\pm0.25}}$ |

**Figure 2: The runtime of different learning methods on the graph Photo.**

**Table 4: The node classification accuracy (%) on a large graph.**

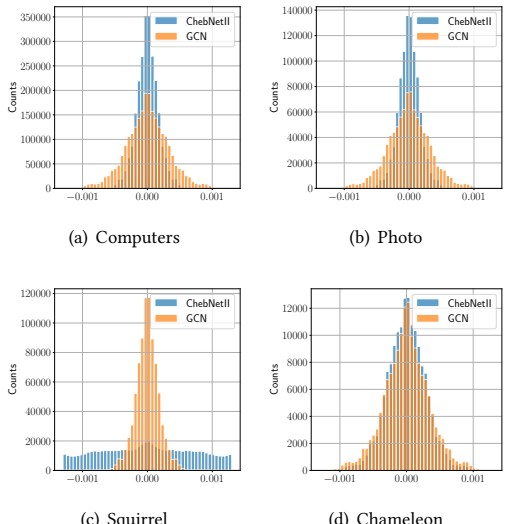

(a) Computers      (b) Photo

(c) Squirrel      (d) Chameleon

**Figure 3: The histogram of $F$'s values for different GNNs.**

ChebNetII, which implies that the $F^*$ of GCN has more non-zero elements and thus has more significant impacts on label prediction. The numerical results in Table 2 can also verify this claim — in most situations, the performance improvements caused by the optimal label transport is significant for GCN but slight for ChebNetII. For heterophilic graphs, learning GCN still leads to Laplacian distributed label transport. However, the distributions corresponding to ChebNetII are diverse — the distribution for Squirrel is long-tailed while that for Chameleon is still Laplacian.

## 4.3 Analytic Experiments

*4.3.1 Robustness to Label Insufficiency Issue.* Our QW loss-based learning method considers the label transport on graphs, whose feasible domain is determined by the observed training labels. The more labels we observed, the smaller the feasible domain is. To demonstrate the robustness of our method to the label insufficiency issue, we evaluate the performance of our method given different amounts of training labels. We train ChebNetII [21] on four graphs by traditional method and our method, respectively. For each graph,

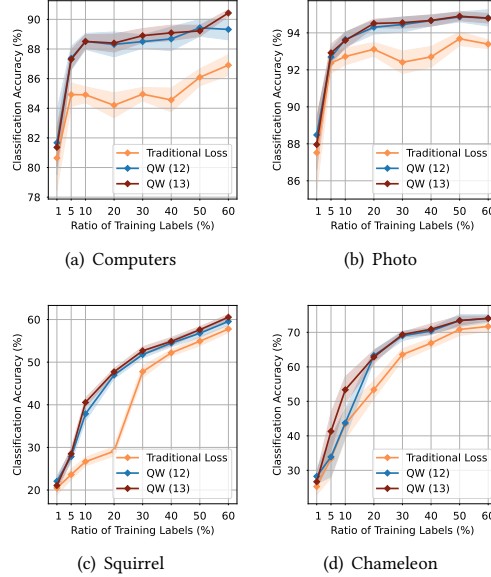

(a) Computers    (b) Photo

(c) Squirrel    (d) Chameleon

**Figure 4: Illustrations of the learning methods' performance given different amounts of labeled nodes.**

**Table 5: Impacts of adjusting edge weights on node classification accuracy (%) when applying the QW loss.**

| Model | | Homophilic | | Heterophilic | |
|---|---|---|---|---|---|
| | | Computers | Photo | Actor | Cornell |
| GCN | $A$ | $\mathbf{88.39_{\pm 0.55}}$ | $\mathbf{93.80_{\pm 0.37}}$ | $30.14_{\pm 0.80}$ | $60.64_{\pm 4.26}$ |
| | $A(F;\xi)$ | $84.35_{\pm 0.46}$ | $91.79_{\pm 0.21}$ | $\mathbf{38.09_{\pm 0.50}}$ | $\mathbf{84.26_{\pm 2.98}}$ |
| ChebNetII | $A$ | $\mathbf{89.52_{\pm 0.54}}$ | $\mathbf{94.84_{\pm 0.37}}$ | $\mathbf{41.37_{\pm 0.67}}$ | $86.38_{\pm 3.19}$ |
| | $A(F;\xi)$ | $89.41_{\pm 0.41}$ | $94.79_{\pm 0.45}$ | $40.74_{\pm 0.80}$ | $\mathbf{86.60_{\pm 2.98}}$ |

we use $K\%$ nodes' labels to train the ChebNetII, where $K \in [1, 60]$, and apply 20% nodes for validation and 20% nodes for testing, respectively, as the above default setting does. Figure 4 shows that our QW loss-based learning method can achieve encouraging performance even when only 20% nodes or fewer are labeled. Additionally, the methods are robust to the selection of solver — we can minimize the QW loss based on (12) or (13), leading to comparable results and outperforming the traditional loss consistently.

*4.3.2 Impacts of Adjusting Edge Weights.* As shown in (17), we can train an MLP to predict edge weights based on the optimal label transport. The ablation study in Table 5 quantitatively show the impacts of adjusting edge weights on the learning results. We can find that for ChebNetII, the two settings provide us with comparable learning results. For GCN, however, learning the model with adjusted edge weights suffers from performance degradation on homophilic graphs while leads to significant improvements on heterophilic graphs. Empirically, it seems that adjusting edge weights based on label transportation helps to improve the learning of simple GNN models on heterophilic graphs.

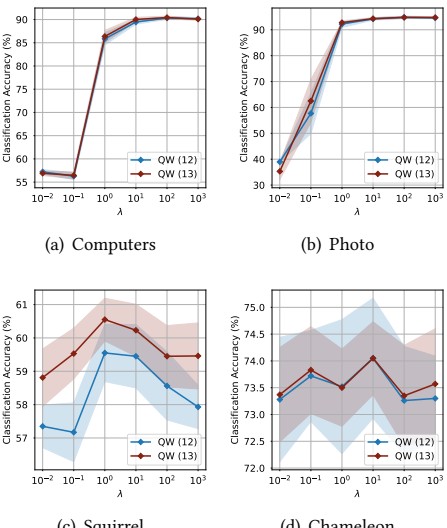

(a) Computers    (b) Photo

(c) Squirrel    (d) Chameleon

**Figure 5: Illustrations of the learning results achieved under different $\lambda$'s.**

*4.3.3 Robustness to Hyperparameters.* The weight of the Bregman divergence term, i.e., $\lambda$, is the key hyperparameter impacting the performance of our learning method. Empirically, when $\lambda$ is too small, the Bregman divergence between the observed labels and their predictions becomes ignorable. Accordingly, the regularizer may be too weak to supervise GNNs' learning properly. On the contrary, when $\lambda$ is too large, the regularizer becomes dominant in the learning objective, and the impact of the label transport becomes weak in both the learning and prediction phases. As a result, it may perform similarly to the traditional method when using a large $\lambda$. We test the robustness of our method to $\lambda$ and show representative results in Figure 5. In particular, our QW loss-based method trains ChebNetII [21] on four graphs. Both Algorithm 1 for (12) and Algorithm 2 for (13) are tested. The $\lambda$ is set in the range from $10^{-2}$ to $10^3$. For homophilic graphs, our learning method achieves stable performance when $\lambda \geq 10$. When $\lambda < 10$, the learning results degrade significantly because of inadequate supervision. Our learning method often obtains the best learning result for heterophilic graphs when $\lambda \in [1, 10]$. These experimental results show that our method is robust to the setting of $\lambda$, and we can set $\lambda$ in a wide range to obtain relatively stable performance.

## 5 CONCLUSION

We have proposed the Quasi-Wasserstein loss for learning graph neural networks. This loss matches well with the non-i.i.d. property of graph-structured data, providing a new strategy to leverage observed node labels in both training and testing phases. Applying the QW loss to learn GNNs improves their performance in various node-level prediction tasks. In the future, we would like to explore the impacts of the optimal label transport on the generalization power of GNNs in theory. Moreover, we plan to modify the QW loss further, developing a new optimization strategy to accelerate its computation.

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

# A THE PROOFS OF THEOREMS

## A.1 Proof of Theorem 1

PROOF. The proof includes four parts:

- **Feasibility.** For $\mu, \gamma \in \text{Range}(S_\mathcal{V})$, we have

$$W_1(\mu, \gamma) = \min_{f \in \Omega(S_\mathcal{V}, \mu, \gamma)} \|\text{diag}(w)f\|_1. \tag{19}$$

Because $\mu, \gamma \in \text{Range}(S_\mathcal{V})$, the feasible domain $\Omega(S_\mathcal{V}, \mu, \gamma)$ is always non-empty and the optimization problem in (19) is always valid.

- **Positivity.** Obviously, the objective in (19) is nonnegative, so that $W_1(\mu, \gamma) \geq 0$, $\forall \mu, \gamma \in \text{Range}(S_\mathcal{V})$. Moreover, let

$$f^* = \arg \min_{f \in \Omega(S_\mathcal{V}, \mu, \gamma)} \|\text{diag}(w)f\|_1.$$

$W_1(\mu, \gamma) = 0$ when $f^* = \mathbf{0}_{|\mathcal{E}|}$, which means $\mu = \gamma$. Therefore, $\forall \mu, \gamma \in \text{Range}(S_\mathcal{V})$, $W_1(\mu, \gamma) \geq 0$ and the equality holds iff $\mu = \gamma$.

- **Symmetry.** Obviously, if $f^*$ is the optimal solution corresponding to $W_1(\mu, \gamma)$, $-f^*$ will be the optimal solution of $W_1(\gamma, \mu)$. Because the edge weight vector is nonnegative, we have $\|\text{diag}(w)f^*\|_1 = \| - \text{diag}(w)f^*\|_1$. As a result, $W_1(\mu, \gamma) = W_1(\gamma, \mu)$.

- **Triangle Inequality.** For $\mu, \gamma, \zeta \in \text{Range}(S_\mathcal{V})$, let

$$W_1(\mu, \gamma) = \min_{f_1 \in \Omega(S_\mathcal{V}, \mu, \gamma)} \|\text{diag}(w)f_1\|_1,$$
$$W_1(\gamma, \zeta) = \min_{f_2 \in \Omega(S_\mathcal{V}, \gamma, \zeta)} \|\text{diag}(w)f_2\|_1.$$

Then, we have

$$\begin{aligned}
&W_1(\mu, \gamma) + W_1(\gamma, \zeta) \\
=& \min_{f_1, f_2} \|\text{diag}(w)f_1\|_1 + \|\text{diag}(w)f_2\|_1 \\
&s.t.\ f_1 \in \Omega(S_\mathcal{V}, \mu, \gamma),\ f_2 \in \Omega(S_\mathcal{V}, \gamma, \zeta) \\
\geq& \min_{f_1, f_2 \in \mathbb{R}^{|\mathcal{E}|}} \|\text{diag}(w)f_1\|_1 + \|\text{diag}(w)f_2\|_1 \\
&s.t.\ S_\mathcal{V}(f_1 + f_2) = \zeta - \mu \\
=& \min_{\tau, \delta \in \mathbb{R}^{|\mathcal{E}|}} \|\text{diag}(w)\tau + \delta\|_1 \\
&\quad\quad\quad + \|\text{diag}(w)\tau - \delta\|_1 \\
&s.t.\ 2S_\mathcal{V}\tau = \zeta - \mu \\
=& \|\text{diag}(w)\hat{\tau} + \hat{\delta}\|_1 + \|\text{diag}(w)\hat{\tau} - \hat{\delta}\|_1 \\
\geq& \|2\text{diag}(w)\hat{\tau}\|_1 \\
\geq& \min_{f \in \Omega(S_\mathcal{V}, \mu, \zeta)} \|\text{diag}(w)f\|_1 \\
=& W_1(\mu, \zeta).
\end{aligned} \tag{20}$$

Here, $\delta := 0.5\text{diag}(w)(f_1 - f_2)$, $\tau := 0.5(f_1 + f_2)$, and

$$\begin{aligned}
\hat{\tau}, \hat{\delta} =& \arg \min_{\tau, \delta \in \mathbb{R}^{|\mathcal{E}|}} \|\text{diag}(w)\tau + \delta\|_1 \\
&\quad\quad\quad + \|\text{diag}(w)\tau - \delta\|_1 \\
&s.t.\ 2S_\mathcal{V}\tau = \zeta - \mu.
\end{aligned}$$

In (20), the first inequality is because the number of constraints is reduced and the feasible domain becomes larger. The second inequality leverages the triangular inequality of $\ell_1$-norm. The third inequality is because $2\hat{\tau}$ is a feasible solution (rather than the optimal solution) corresponding to $W_1(\mu, \zeta)$.

Replacing $\mathcal{V}$ to a subset of nodes $\mathcal{V}' \subset \mathcal{V}$, we obtain a partial Wasserstein distance $W_1^{(P)}$ defined on the graph. Based on the same steps, we can prove that $W_1^{(P)}$ is a valid metric in $\text{Range}(S_{\mathcal{V}'})$. □

## A.2 The Proof of Theorem 2

PROOF. Denote $\mathcal{V}_U = \mathcal{V} \setminus \mathcal{V}'$. Based on the shrinkage of the feasible domain, we have

$$\begin{aligned}
W_1(\mu, \gamma) =& \min_{f \in \Omega(S_\mathcal{V}, \mu, \gamma)} \|\text{diag}(w)f\|_1 \\
=& \min_{f \in \Omega(S_{\mathcal{V}'}, \mu_{\mathcal{V}'}, \gamma_{\mathcal{V}'}) \cap \Omega(S_{\mathcal{V}_U}, \mu_{\mathcal{V}_U}, \gamma_{\mathcal{V}_U})} \|\text{diag}(w)f\|_1 \\
\geq& \min_{f \in \Omega(S_{\mathcal{V}'}, \mu_{\mathcal{V}'}, \gamma_{\mathcal{V}'})} \|\text{diag}(w)f\|_1 = W_1^{(P)}(\mu_{\mathcal{V}'}, \gamma_{\mathcal{V}'}).
\end{aligned}$$

The second inequality in (9) can be proven in the same way. The nonnegativeness is based on the metricity. □

# B EXPERIMENTAL DETAILS

## B.1 Baseline Implementations and Experimental Settings

All baseline models are implemented using the code released by the respective authors, as provided below.

- **GCN, GAT, APPNP, and BernNet:** https://github.com/ivam-he/BernNet
- **ChebNetII:** https://github.com/ivam-he/ChebNetII
- **GCN-LPA:** https://github.com/hwwang55/GCN-LPA

For GCN, GAT, GIN, GraphSAGE, and APPNP, we search the learning rate over the range of $\{0.001, 0.002, 0.01, 0.05\}$ and the weight decay over the range of $\{0.0, 0.0005\}$. For APPNP, we search its key hyperparameter $\alpha$ over $\{0.1, 0.2, 0.5, 0.9\}$. For BernNet and ChebNetII, we used the hyperparameters provided by the original papers [21, 22]. For GCN-LPA, we apply a two-layer GCN associated with five LPA iteration layers, which follows the settings in [46]. We utilize the same datasets and data partitioning as BernNet [22] and ChebNetII [21] in our experiments.

## B.2 Hyperparameter Settings

For all GNN methods, we modify their architectures according to Algorithms 1 and 2 and learn the models through the QW loss. The key hyperparameters and their search spaces are shown below: $A_F$ indicates whether the optimal label transport is involved in the adjustment of edge weights, which is set to True or False. $\lambda$ is the weights of Bergman divergence, whose search space is $\{10^{-2}, 10^{-1}, ..., 10^3\}$. $lr_F$ and $L_F$ denote the learning rate and weight decay for the label transport $F$ and MLP-based edge weight predictor. We search for parameter $lr_F$ over the range of $\{0.001, 0.002, 0.01, 0.05\}$ and parameter $L_F$ over the range of $\{0.0, 0.0005\}$.

