# OpenReview forum: "A Quasi-Wasserstein Loss for Learning Graph Neural Networks"
_ACM.org/TheWebConf/2024/Conference — TheWebConf24_

### Official Review · Reviewer_cukF · 2023-11-22

**Novelty:** 6
**Technical Quality:** 5

**Review:**

This work proposes a *Quasi-Wasserstein (QW) loss* to better model instances
where node labels are not i.i.d. This approach is inspired by the well-studied
optimal transport problem. The authors do a good job motivating this new loss
function, drawing connections to previous works, and give a comprehensive set
of experiments.

**Strengths**
- The relaxation of the flow matrix constraint to a Bregman divergence-based
  regularizer in Section 3.3.1 makes this method more approachable for
  practitioners.
- The experiments are compelling and well-documented. The authors consider many
  academic homophilic and heterophilic grpahs, and they compare traditional
  node-level loss with their class-based QW loss for spatial GNNs (e.g., GCN and
  GraphSAGE) and spectral GNNs (BernNet and ChebNetII).
- Overall, the paper is sufficiently mathematically technical and
  well-motivated.

**Weaknesses**
- [line 481] To make Table 2 easier to read in isolation, consider replacing
  (2) in the Method column with "CE" for cross entropy.
- [line 424] Since QW and cross entropy are different loss functions, it would
  be helpful to record in the experiments how the number of trainable
  parameters differs in these optimization problems, for each algorithm and
  dataset. If QW uses noticeably more parameters, then some of the improvements
  could potentially come from over-parameterization and better training dynamics.

**Typos and suggestions**
- [line 344] suggestion: Replace "The Validness as A Metric" by "Metric
  Property"
- [line 352] suggestion: Replace "The Monotonicity" by "Monotonicity"
- [line 727] typo: "ti" --> "it"

**Questions:**

See comment above about the number of trainable parameters for each (model, method, graph) instance in the experiments.

**Reviewer Confidence:**

2: The reviewer is willing to defend the evaluation, but it is likely that the reviewer did not understand parts of the paper

**Scope:**

4: The work is relevant to the Web and to the track, and is of broad interest to the community

---

### Official Review · Reviewer_3Eac · 2023-11-22

**Novelty:** 4
**Technical Quality:** 5

**Review:**

The paper introduces a new loss for learning GNNs, which considers the non-iid nature of graph data when predicting. The proposed loss is general and could apply to various GNN tasks and datasets. Extensive results well demonstrate the effectiveness of the proposed method over traditional loss.

Pros:
1. The studied motivation is clear and reasonable for graph data.
2. The proposed loss is effective over traditional cross-entropy loss.

Cons:
1. The idea of considering the relationship between labels has been studied in the GMNN. What are the major differences between the proposed method with GMNN in terms of motivation and performance?
[1] Qu M, Bengio Y, Tang J. Gmnn: Graph markov neural networks[C]//International conference on machine learning. PMLR, 2019: 5241-5250.

**Questions:**

The idea of considering the relationship between labels has been studied in the GMNN. What are the major differences between the proposed method with GMNN in terms of motivation and performance?

**Reviewer Confidence:**

3: The reviewer is confident but not certain that the evaluation is correct

**Scope:**

4: The work is relevant to the Web and to the track, and is of broad interest to the community

---

### Official Review · Reviewer_GdS3 · 2023-11-23

**Novelty:** 5
**Technical Quality:** 4

**Review:**

This paper presents a new metric, called quasi-Wasserstein loss, for graph neural networks.
By noticing that the nodes in a graph-structured data are not independent, while the traditional risk often has an i.i.d. assumption, it builds a new loss for graph neural networks based on optimal transport optimization.
Some experimental results are demonstrated to show improvement over baseline metrics.

Quality: Overall, this paper is good. The experimental evaluations can be improved.

On lines 671-672, the split ratio is 60% for training, 20% for validation, and 20% for test.
The comparison appears not so fair. The original GCN and GAT papers, though using (2) as objective functions, are for semi-supervised learning. They used a small fraction of nodes for training, which is far less than 60% for training used in this paper.
Therefore, it  would be better to compare with these methods  in the original  setting so that readers may get a better assessment regarding the improvement.



Clarity: The paper is clearly presented.

Originality: The novelty appears good.

Significance: The topic is important and has significance in real applications.

**Questions:**

Generally, the nodes have correlations due to the connections in the graph-structured data as discussed in the paper. So, it is expected that on most graph datasets, the new loss function, which is dedicated to resolve the non-i.i.d. issue, would have improvement over the methods using classic  loss functions. However, the experiment results show that on many datasets little improvement could be obtained. Why did this happen? Do you have an explanation?

**Reviewer Confidence:**

3: The reviewer is confident but not certain that the evaluation is correct

**Scope:**

4: The work is relevant to the Web and to the track, and is of broad interest to the community

---

### Official Review · Reviewer_8EjN · 2023-11-27

**Novelty:** 6
**Technical Quality:** 6

**Review:**

**Strengths**

1. The QW loss is a new optimal transport-based loss that addresses the inconsistency issue in traditional GNN learning. It avoids the questionable independent and identically distributed (i.i.d.) assumption by considering the labels and estimations of graph nodes jointly, making it a valid metric for node labels on graphs.

2. The QW loss introduces a novel approach to combine node embeddings with label information in both training and testing phases. It uses Bregman divergence-based algorithms for learning and combines the GNN output with a residual component from optimal label transport for predictions, leading to a new transductive prediction paradigm.
3. Experiments have shown that applying the QW loss to various GNNs enhances their performance in node-level classification and regression tasks.



**Weaknesses**

1. A notable limitation of the QW loss is the increased computational demand associated with the implementation of optimal transport-based methods, especially for Algorithm 3.
2. The authors did not compare the runtimes on large-scale graphs. The scalability of the QW loss in handling extremely large graphs might be a concern.
3. The efficacy of the learnable edge-weight module, a component of the QW loss framework, appears to be somewhat constrained.
4. A minor typo ('ti') was noted around line 727.

**Questions:**

1. Could the authors provide a comparative analysis between Algorithm 1 and Algorithm 2 on node classification accuracy? An illustrative table in the appendix detailing this comparison would be greatly beneficial for a clearer understanding of the relative strengths and weaknesses of each algorithm.
2. In cases of homophilic graphs, what potential reasons might the authors suggest for the observed limited improvement when adjusting edge weights? Insight into this aspect could help in better understanding the applicability and limitations of the QW loss in different graph structures.
3. Are there any practical recommendations that the authors could offer to mitigate the risk of overfitting associated with the QW loss? Specifically, are there approaches other than training both Algorithms 2 and 3 and selecting based on performance?

**Reviewer Confidence:**

4: The reviewer is certain that the evaluation is correct and very familiar with the relevant literature

**Scope:**

4: The work is relevant to the Web and to the track, and is of broad interest to the community

---

### Decision · Program_Chairs · 2024-01-22

**Decision:**

Accept

**Comment:**

The reviewers are in agreement that this is a good paper for this conference, with some room for improvement in the experiments and presentation.